

# A solvable model for graph state decoherence dynamics

**Jérôme Houdayer[1][*], Haggai Landa[2] and Grégoire Misguich[1]**

**1** Université Paris-Saclay, CNRS, CEA, Institut de physique théorique,
91191 Gif-sur-Yvette, France
**2** IBM Quantum, IBM Research – Israel, Haifa University Campus,
Mount Carmel, Haifa 31905, Israel

[*] jerome.houdayer@ipht.fr

## Abstract

We present an exactly solvable toy model for the continuous dissipative dynamics of permutation-invariant graph states of $N$ qubits. Such states are locally equivalent to an $N$-qubit Greenberger-Horne-Zeilinger (GHZ) state, a fundamental resource in many quantum information processing setups. We focus on the time evolution of the state governed by a Lindblad master equation with the three standard single-qubit jump operators, the Hamiltonian part being set to zero. Deriving analytic expressions for the expectation values of observables expanded in the Pauli basis at all times, we analyze the nontrivial intermediate-time dynamics. Using a numerical solver based on matrix product operators, we simulate the time evolution for systems with up to 64 qubits and verify a numerically exact agreement with the analytical results. We find that the evolution of the operator space entanglement entropy of a bipartition of the system manifests a plateau whose duration increases logarithmically with the number of qubits, whereas all Pauli-operator products have expectation values decaying at most in constant time.

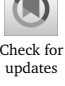

# 1   Introduction

Graph states were introduced by Briegel and Raussendorf in 2001 [1] as special entangled states of $N$ qubits. These states with multipartite entanglement play an important role in quantum information theory because they can be employed as a resource in a measurement-based quantum computation [2,3], and they can be used in error correction codes [4] and for quantum communications [5]. In particular, permutation-invariant graph states [6], which are locally equivalent to an $N$-qubit Greenberger-Horne-Zeilinger (GHZ) state, are the subject of extensive research [7–10] and their creation and characterization serve as one of a few standard benchmarks of quantum computation hardware [11–18]. The use of graph states for information processing in current quantum devices will inevitably have to face uncontrolled decoherence processes and some aspects of graph state entanglement under the presence of decoherence have already been investigated [19–22]. Most of these previous works focused on discrete evolutions of the density matrix via completely positive maps (noisy channels).

In this work, we introduce and discuss a model of a graph state realized with qubits (or spin-one-half particles), which decoheres continuously in time as described by a Lindblad master equation for the density operator [23,24]. We account for the three most prevalent local jump operators (dissipators); Two jump terms describe the incoherent transitions from $|0\rangle$ to $|1\rangle$ (and vice versa), and the third one is the so-called dephasing term. The initial state is a pure (graph) state, and it evolves into a mixed state under the action of the dissipation. Although it is a many-body problem, the structure of the model is simple enough that the expectation values of any observable can here be computed exactly by solving the equations of motion for the expectation values of products of Pauli matrices.

We complement our analytic treatment with the use of a numerical Lindblad solver [25, 26], which is internally based on the C++ ITensor library [27] (see also [28] for a review on available numerical methods for this type of problem). In the solver, the state of the system – a many-body density matrix $\rho$ – is stored in the form a matrix-product operator (MPO). Since $\rho$ is in general a matrix of size $2^N \times 2^N$, a brute-force numerical simulation of the Lindblad dynamics generally becomes very demanding beyond a dozen of qubits. Taking advantage of the fact that the states produced along the time evolution are only mildly correlated in the present model, the MPO approach allows us to reach a very high accuracy with modest computing resources (i.e. low MPO bond dimension) even with as many as $N = 64$ qubits. Other numerical approaches would also be efficient in the context of the current setup [6,29].

The presented model can be viewed as a toy model illustrating the basic mechanisms at play and allowing one to gain an understanding of the dominant dynamical behavior in similar setups. It may also be used as a starting point for more realistic studies with different graph states and some Hamiltonian terms competing with the dissipators. With our numerical approach, we are able to calculate global quantities that are not immediately accessible analytically, and observe an interesting scaling with size of bipartite correlations in the system.

In the next section, we present the model. In Sec. 3, we show how to compute exactly the evolution of the expectation value of any product of Pauli matrices in this model. From this result, one can then obtain the expectation value of any observable as a function of time. In Sec. 4, we then compare these analytical results with numerical simulations based on a matrix product operator (MPO) representation of the density matrix. Sec. 5 provides some concluding remarks.

## 2  Notations and definition of the model

We consider a system composed of $N$ qubits (with basis states $|0\rangle = |\uparrow\rangle$ and $|1\rangle = |\downarrow\rangle$), indexed with $i = 1 \ldots N$. We denote the usual Pauli operators by $\sigma^x$, $\sigma^y$ and $\sigma^z$, or alternatively by $X$, $Y$ and $Z$. Additionally, we define $\sigma^{\pm} = \frac{1}{2}(\sigma^x \pm i\sigma^y)$.

For completeness, we start by recalling the definition of a graph state. A graph state is an entangled state that can be produced by the symmetrical two-qubit gate "controlled-Z" (CZ), which is defined by its matrix

$$CZ = \begin{pmatrix} 1 & 0 & 0 & 0 \\ 0 & 1 & 0 & 0 \\ 0 & 0 & 1 & 0 \\ 0 & 0 & 0 & -1 \end{pmatrix}, \tag{1}$$

in the basis $\{|00|\rangle, |10|\rangle, |01|\rangle, |11|\rangle\}$. Given an undirected graph $G(V, E)$ where $V = 1 \ldots N$ represents the qubits and $E \subset V \times V$ is the set of edges, the corresponding graph state $|g\rangle$ is defined by

$$|g\rangle = \prod_{(i,j) \in E} CZ(i,j)|++\cdots+\rangle, \tag{2}$$

where $|+\rangle = \frac{1}{\sqrt{2}}(|0\rangle + |1\rangle)$. It is well known [1], that the graph state $|g\rangle$ is characterized by its stabilizers

$$S_i = \sigma_i^x \prod_{j|(i,j) \in E} \sigma_j^z, \tag{3}$$

through

$$S_i|g\rangle = |g\rangle. \tag{4}$$

In the present study, we focus on the case where the system is *invariant under any permutation of the N qubits*. We start at $t = 0$ from the only non-trivial fully symmetrical graph state, that is the graph state associated to the complete graph. A complete graph is a graph where all possible edges are present, so that each vertex is linked to the $N - 1$ other vertices. Our initial state (at $t = 0$) is thus given by

$$|g\rangle = \prod_{i<j} CZ(i,j)|++\cdots+\rangle. \tag{5}$$

As a side remark, we note that, thanks to the transformation rules of graph states under the action of local Clifford (LC) gates [20], the complete graph is LC-equivalent to the star graph.[1] In turn, the star graph state can be transformed into the $N$-qubit Greenberger-Horne-Zeilinger (GHZ) state [30] by application of Hadamard gates to all qubits except the center of the star. The complete graph is thus LC-equivalent to the GHZ state.

We consider a time evolution generated by a Lindblad equation (see for example [31]) where the Hamiltonian part is set to zero. The state of the system is described by its density matrix $\rho$ whose time evolution is given by

$$\frac{\partial}{\partial t}\rho = \mathcal{D}[\rho], \tag{6}$$

where $\mathcal{D}$ is the dissipator, a linear superoperator acting on $\rho$. We consider three possible terms

---

[1]The star graph has a central vertex connected to all the other vertices.

in the dissipator $\mathcal{D} = \mathcal{D}_0 + \mathcal{D}_1 + \mathcal{D}_2$. They are given by

$$\mathcal{D}_0[\rho] = g_0 \sum_i \left( \sigma_i^+ \rho \sigma_i^- - \frac{1}{2}\{\sigma_i^- \sigma_i^+, \rho\} \right), \tag{7}$$

$$\mathcal{D}_1[\rho] = g_1 \sum_i \left( \sigma_i^- \rho \sigma_i^+ - \frac{1}{2}\{\sigma_i^+ \sigma_i^-, \rho\} \right), \tag{8}$$

$$\mathcal{D}_2[\rho] = g_2 \sum_i \left( \sigma_i^z \rho \sigma_i^z - \rho \right), \tag{9}$$

where $\{,\}$ is the anticommutator.[2] $\mathcal{D}_0$ (resp. $\mathcal{D}_1$) then corresponds to an incoherent transition toward the state $|0\rangle$ (resp. $|1\rangle$). $\mathcal{D}_2$ corresponds to a dephasing in the $xy$-plane.

## 3 Closed-form observable dynamics

In this section, we derive some exact results for the time evolution of a large class of observables.

### 3.1 Observable dynamics

To compute the evolution of the mean value of a given (time-independent) observable $O$, we start from the fact that $\langle O \rangle = \mathrm{Tr}(\rho O)$ and we use Eq. 6 to get

$$\begin{aligned}
\frac{\partial}{\partial t}\langle O \rangle = \frac{\partial}{\partial t}\mathrm{Tr}(\rho O) &= \mathrm{Tr}\left( \frac{\partial \rho}{\partial t} O \right) \\
&= \mathrm{Tr}(\mathcal{D}[\rho]O) \\
&= \mathrm{Tr}(\mathcal{D}_0[\rho]O) + \mathrm{Tr}(\mathcal{D}_1[\rho]O) + \mathrm{Tr}(\mathcal{D}_2[\rho]O).
\end{aligned} \tag{10}$$

First let us consider $\mathcal{D}_0$

$$\begin{aligned}
\mathrm{Tr}(\mathcal{D}_0[\rho]O) &= g_0 \sum_i \mathrm{Tr}\left[ (\sigma_i^+ \rho \sigma_i^- - \frac{1}{2}\{\sigma_i^- \sigma_i^+, \rho\})O \right] \\
&= g_0 \sum_i \mathrm{Tr}\left[ \rho \left( \sigma_i^- O \sigma_i^+ - \frac{1}{2}\{\sigma_i^- \sigma_i^+, O\} \right) \right] \\
&= g_0 \sum_i \langle \Lambda_0^i[O] \rangle,
\end{aligned} \tag{11}$$

where the superoperator $\Lambda_0^i$ is given by

$$\Lambda_0^i[O] = \sigma_i^- O \sigma_i^+ - \frac{1}{2}\{\sigma_i^- \sigma_i^+, O\}. \tag{12}$$

Similarly, we have

$$\mathrm{Tr}(\mathcal{D}_1[\rho]O) = g_1 \sum_i \langle \Lambda_1^i[O] \rangle, \tag{13}$$

$$\mathrm{Tr}(\mathcal{D}_2[\rho]O) = g_2 \sum_i \langle \Lambda_2^i[O] \rangle, \tag{14}$$

---

[2] $\{A, B\} = AB + BA$.

with

$$\Lambda_1^i[O] = \sigma_i^+ O \sigma_i^- - \frac{1}{2}\{\sigma_i^+ \sigma_i^-, O\}, \tag{15}$$

$$\Lambda_2^i[O] = \sigma_i^z O \sigma_i^z - O. \tag{16}$$

It is clear that if $O$ does not operate on qubit $i$ then $\Lambda_*^i[O] = 0$. Moreover, for an operator $O_i$ acting on qubit $i$ only, $\Lambda_*^i[O_i]$ commutes with operators acting on the other qubits. The nontrivial action of the $\Lambda_*^i$ can then be summarized by the following relations:

$$\Lambda_0[\sigma^x] = -\frac{1}{2}\sigma^x, \qquad \Lambda_1[\sigma^x] = -\frac{1}{2}\sigma^x, \qquad \Lambda_2[\sigma^x] = -2\sigma^x,$$

$$\Lambda_0[\sigma^y] = -\frac{1}{2}\sigma^y, \qquad \Lambda_1[\sigma^y] = -\frac{1}{2}\sigma^y, \qquad \Lambda_2[\sigma^y] = -2\sigma^y,$$

$$\Lambda_0[\sigma^z] = 1 - \sigma^z, \qquad \Lambda_1[\sigma^z] = -1 - \sigma^z, \qquad \Lambda_2[\sigma^z] = 0,$$

where the qubit index of the Pauli operators are identical in the l.h.s and r.h.s and have been omitted for brevity.

## 3.2 Observables at $t = 0$

To lighten the notations, we will write $X_i$ instead of $\sigma_i^x$ and likewise for $Y$ and $Z$. As our system is invariant under any permutation of the qubits, specific indices are irrelevant. More generally, we note that $\langle X^n Y^m Z^l \rangle = \langle \sigma_1^x \ldots \sigma_n^x \sigma_{n+1}^y \ldots \sigma_{n+m}^y \sigma_{n+m+1}^z \ldots \sigma_{n+m+l}^z \rangle$ which is independent of the actual order of the operators or the specific indices as long as they are all different. When indices are necessary, we will add them, for example $\langle X_1 Z_1 Y^2 Z \rangle$ is the same as $\langle X_1 Z_1 Y_2 Y_3 Z_4 \rangle$. Likewise, $\langle (X_i Z_i)^2 (Y_j X_j)^2 Z \rangle$ has the same value as $\langle X_1 Z_1 X_2 Z_2 Y_3 X_3 Y_4 X_4 Z_5 \rangle$.

To compute the expectation value of a product of Pauli operators at $t = 0$, that is on the complete graph state $|g\rangle$, we start with two remarks. First the stabilizer $S = X Z^{N-1}$ leaves $|g\rangle$ unchanged (see Eqs. 3 and 4) so that

$$\langle X Z^{N-1} \rangle = \langle g | X Z^{N-1} | g \rangle = \langle g | g \rangle = 1. \tag{17}$$

Second, since $Z$ commutes with $CZ$ and since $CZ^2 = 1$, we have for $n > 0$

$$\begin{aligned}
\langle Z^n \rangle &= \langle g | Z^n | g \rangle \\
&= \langle + \cdots + | Z^n | + \cdots + \rangle \\
&= \langle + \cdots + | - \cdots - + \cdots + \rangle \\
&= 0.
\end{aligned} \tag{18}$$

We start by computing observables of the form $\langle X^n Z^l \rangle$. To do this, we introduce the stabilizer at one of the indices of the $X$. So for $n > 0$

$$\begin{aligned}
\langle X^n Z^l \rangle &= \langle X_1 X^{n-1} Z^l X_1 Z^{N-1} \rangle \\
&= \langle (X_i Z_i)^{n-1} Z^{N-l-n} \rangle \\
&= (-i)^{n-1} \langle Y^{n-1} Z^{N-l-n} \rangle \\
&= (-1)^{\frac{n-1}{2}} \langle Y^{n-1} Z^{N-l-n} \rangle.
\end{aligned} \tag{19}$$

To be real, this last expression above must be 0 when $n$ is even. We can do the same for $\langle Y^m Z^l \rangle$, which gives for $m > 0$

$$\begin{aligned}
\langle Y^m Z^l \rangle &= \langle Y_1 Y^{m-1} Z^l X_1 Z^{N-1} \rangle \\
&= \langle Y_1 X_1 (Y_i Z_i)^{m-1} Z^{N-l-m} \rangle \\
&= i^{m-2} \langle X^{m-1} Z^{N-l-m+1} \rangle \\
&= (-1)^{\frac{m}{2}-1} \langle X^{m-1} Z^{N-l-m+1} \rangle.
\end{aligned} \tag{20}$$

And again this must be 0 if $m$ is odd. Substituting Eq. 19 in Eq. 20, we can conclude that for even $m$

$$\langle Y^m Z^l \rangle = \langle Y^{m-2} Z^l \rangle = \langle Z^l \rangle, \tag{21}$$

which is 0 if $l > 0$ and 1 otherwise. We can also finish the computation for $X$ for odd $n$

$$\begin{aligned}
\langle X^n Z^l \rangle &= (-1)^{\frac{n-1}{2}} \langle Y^{n-1} Z^{N-l-n} \rangle \\
&= (-1)^{\frac{n-1}{2}} \langle Z^{N-l-n} \rangle,
\end{aligned} \tag{22}$$

which is 1 if $n + l = N$, and zero otherwise. The last product we have not yet computed is the general one $\langle X^n Y^m Z^l \rangle$ with $n > 0$ and $m > 0$.

$$\begin{aligned}
\langle X^n Y^m Z^l \rangle &= \langle X_1 X^{n-1} Y^m Z^l X_1 Z^{N-1} \rangle \\
&= \langle (X_i Z_i)^{n-1} (Y_j Z_j)^m Z^{N-n-m-l} \rangle \\
&= i^{m-n+1} \langle X^m Y^{n-1} Z^{N-n-m-l} \rangle \\
&= (-1)^{\frac{m-n+1}{2}} \langle X^m Y^{n-1} Z^{N-n-m-l} \rangle.
\end{aligned} \tag{23}$$

This can be nonzero only if $n + m$ is odd. But if $n = 1$ there are no $Y$ left and this is zero according to Eq. 22. And if $n > 1$, the right-hand side can be nonzero only if $n + m - 1$ is odd (thus $n + m$ is even) which we have already excluded. Thus, $\langle X^n Y^m Z^l \rangle = 0$ if $n > 0$ and $m > 0$.

To summarize, only the following products of Pauli operators have nonzero mean values in the complete graph state:

$$\langle X^{2n+1} Z^{N-2n-1} \rangle = (-1)^n, \qquad \text{and} \qquad \langle Y^{2n} \rangle = 1. \tag{24}$$

## 3.3 Solution of the equations of motion

We start with an example to show how the equation of motion leads to some differential equations. Here we consider $\langle XZ \rangle$ in the case where only $g_0$ is not zero.

$$\begin{aligned}
\frac{\partial}{\partial t} \langle XZ \rangle &= \frac{\partial}{\partial t} \langle X_1 Z_2 \rangle \\
&= \text{Tr}[\mathcal{D}_0[\rho](X_1 Z_2)] \\
&= g_0 \sum_i \langle \Lambda_0^i [X_1 Z_2] \rangle \\
&= g_0 \left( \langle \Lambda_0^1 [X_1] Z_2 \rangle + \langle X_1 \Lambda_0^2 [Z_2] \rangle \right) \\
&= g_0 \left( -\frac{1}{2} \langle X_1 Z_2 \rangle + \langle X_1 (1 - Z_2) \rangle \right) \\
&= g_0 \left( -\frac{3}{2} \langle XZ \rangle + \langle X \rangle \right).
\end{aligned} \tag{25}$$

Now the general formula for $\frac{\partial}{\partial t} \langle X^n Y^m Z^l \rangle$ and all dissipators: each $X$ or $Y$ gives a term $(-g_0/2 - g_1/2 - 2g_2) \langle X^n Y^m Z^l \rangle$ and each $Z$ results in two terms

$$(-g_0 - g_1) \langle X^n Y^m Z^l \rangle, \qquad (g_0 - g_1) \langle X^n Y^m Z^{l-1} \rangle. \tag{26}$$

Globally, we obtain for $l > 0$

$$\frac{\partial}{\partial t} \langle X^n Y^m Z^l \rangle = -(\alpha(n+m) + \beta l) \langle X^n Y^m Z^l \rangle + \gamma l \langle X^n Y^m Z^{l-1} \rangle, \tag{27}$$

$$\frac{\partial}{\partial t} \langle X^n Y^m \rangle = -\alpha(n+m) \langle X^n Y^m \rangle, \tag{28}$$

where

$$\alpha = \frac{g_0 + g_1}{2} + 2g_2, \qquad \beta = g_0 + g_1, \qquad \gamma = g_0 - g_1. \tag{29}$$

Here, $\alpha$ is the rate of dephasing (decoherence) associated to $X$ and $Y$ (with its inverse equal to the characteristic $T_2$ timescale), $\beta$ is the decay parameter associated to $Z$ – the inverse of the relaxation time $T_1$, $\gamma$ is the global drive towards the steady state, and $\gamma/\beta$ determines the thermal steady state population (value of $\langle Z \rangle$) reached in the limit of large time.[3] In the case where $\gamma = 0$ (that is $g_0 = g_1$), all observables have a simple exponential decay.

These equations can be solved by recursion starting at $l = 0$ using the initial conditions of the previous section. Indeed, at $l = 0$ (that is no $Z$), we get

$$\langle Y^{2n} \rangle = e^{-2\alpha n t}, \tag{30}$$

and all the others products without $Z$ give zero because they start at 0 and stay there. Now we can increase $l$ and get

$$\langle Y^{2n} Z^l \rangle = \left( \frac{\gamma}{\beta} \left( 1 - e^{-\beta t} \right) \right)^l e^{-2\alpha n t}. \tag{31}$$

Note that if $\gamma = 0$ or $\gamma = \beta = 0$ then $\langle Y^{2n} Z^l \rangle = 0$ for $l > 0$. Finally for $\langle X^{2n+1} Z^{N-2n-1} \rangle$, the equations are directly solved and we obtain

$$\langle X^{2n+1} Z^{N-2n-1} \rangle = (-1)^n e^{-((2n+1)(\alpha-\beta)+N\beta)t}, \tag{32}$$

with all the others being identically zero.

It is interesting to note that the stabilizers that characterize the initial graph state decay very rapidly, i.e. with a timescale inversely proportional to the size of the system. This reflects some relative fragility of the graph state correlations in the present dissipative context, and may be related to the extensive number of neighbors in the initial complete graph.

### 3.4 Reduced density matrices

The two-qubit density matrix can be obtained from the two-point correlations computed previously. Writing $z_\pm = 1 \pm \langle Z \rangle = 1 \pm \frac{\gamma}{\beta} \left( 1 - e^{-\beta t} \right)$ (see Eq. 31 with $n = 0$ and $l = 1$) and $y^2 = \langle YY \rangle = e^{-2\alpha t}$ (see Eq. 30 with $n = 1$), this matrix reads (for $N > 2$ only)

$$\rho_2 = \frac{1}{4} \begin{pmatrix} z_+^2 & 0 & 0 & -y^2 \\ 0 & z_+ z_- & y^2 & 0 \\ 0 & y^2 & z_+ z_- & 0 \\ -y^2 & 0 & 0 & z_-^2 \end{pmatrix}. \tag{33}$$

Likewise for 3 qubits and $N > 3$, we have

$$\rho_3 = \frac{1}{8} \begin{pmatrix} z_+^3 & 0 & 0 & -y^2 z_+ & 0 & -y^2 z_+ & -y^2 z_+ & 0 \\ 0 & z_- z_+^2 & y^2 z_+ & 0 & y^2 z_+ & 0 & 0 & -y^2 z_- \\ 0 & y^2 z_+ & z_- z_+^2 & 0 & y^2 z_+ & 0 & 0 & -y^2 z_- \\ -y^2 z_+ & 0 & 0 & z_-^2 z_+ & 0 & y^2 z_- & y^2 z_- & 0 \\ 0 & y^2 z_+ & y^2 z_+ & 0 & z_- z_+^2 & 0 & 0 & -y^2 z_- \\ -y^2 z_+ & 0 & 0 & y^2 z_- & 0 & z_-^2 z_+ & y^2 z_- & 0 \\ -y^2 z_+ & 0 & 0 & y^2 z_- & 0 & y^2 z_- & z_-^2 z_+ & 0 \\ 0 & -y^2 z_- & -y^2 z_- & 0 & -y^2 z_- & 0 & 0 & z_-^3 \end{pmatrix}. \tag{34}$$

---

[3]In the long time limit and for $n = 0$ and $l = 1$ the Eq. 31 below gives $\langle Z \rangle_{t \gg \beta} = \frac{\gamma}{\beta}$ which is the thermal equilibrium value if we introduce a local Hamiltonian $H = \Delta \sum_i Z_i$ and a temperature $T$ satisfying $\tanh(\Delta/(k_B T)) = \frac{\gamma}{\beta}$.

More generally, reduced density matrices for larger subsystem can be obtained by noting that each matrix element is the expectation value of a product of $N$ operators which are of the type $(Z_i+1)/2$ (if the matrix element connects two states where $Z_i = 1$), $(1-Z_i)/2$ (matrix element between two states where $Z_i = -1$), or $\sigma_i^{\pm}$ (matrix element between two states with opposite $Z_i$).

Since $\rho_2$ can be decomposed as

$$\rho_2 = \frac{1}{4}\begin{pmatrix} z_+ & 0 \\ 0 & z_- \end{pmatrix} \otimes \begin{pmatrix} z_+ & 0 \\ 0 & z_- \end{pmatrix} + \frac{1}{4}\begin{pmatrix} 0 & -iy \\ iy & 0 \end{pmatrix} \otimes \begin{pmatrix} 0 & -iy \\ iy & 0 \end{pmatrix}, \tag{35}$$

it is easy to see that it is separable (at all times). We also checked that $\rho_3$ and $\rho_4$ (not shown) show no negativity, independent of the time $t$.

It is in fact a well known property of GHZ states [20] that the reduced density matrices are separable, even though the system has some global multipartite entanglement. In the present model, the reduced density matrices are thus all separable at $t = 0$. In addition, since the dynamics is only due to single qubit disspators the reduced density matrices must remain separable at all times (unless one considers the system *globally* (all qubits)).

# 4 Computational results

The possibility to simulate quantum open systems by representing the density matrix as a (one-dimensional) tensor network was originally discussed in Refs. [32,33]. In the present work, we use a representation of $\rho$ as an MPO, closely related to what was done by Prosen and Žnidarič in [34]. In our implementation, the density matrix $\rho$ is encoded in a vectorized form (denoted by $|\rho\rangle\rangle$) as an ITensor matrix-product state (MPS) over an Hilbert space with dimension 4 at each site.[4] More details on this representation can be found in App. A. Concerning the implementation of the time evolution, the first step amounts to expressing the Lindbladian superoperator $\mathcal{L}$ as a (super-) MPO that can act on a vectorized $|\rho\rangle\rangle$. Next, one constructs a (super-) MPO approximation of the exponential $\exp(\tau\mathcal{L})$ associated to a small time step $\tau$. This approximation is based on an extension of the $W^{\mathrm{II}}$ scheme proposed in Ref. [35], and it allows for long-range interactions. In the scheme we use (see appendix A of Ref. [36]), the Trotter error scales as $\mathcal{O}(\tau^5)$. All the simulations used a time step $\tau = 0.004$. The truncation in the MPO was carried out with a maximum discarded weight of $10^{-16}$.

As a pure state, the initial state can be written exactly has a matrix-product state with bond dimension equal to 2, and as a density matrix $\rho$, it can be represented exactly by an MPO of bond dimension equal to 4. It turns out that the local dissipation terms of the model do not increase the required bond dimension. Note however that due to small numerical errors the bond dimension was sometimes observed to be above 4 in the simulations (but never exceeding 15).

## 4.1 Dissipative dynamics of observables

In this section, we present the dynamics of Pauli observables calculated from the analytic expressions of Eqs. 30-32, together with numerical simulation results from the MPO solver. App. A gives more details on the numerical simulations. To explore the parameter space, we studied five representative cases varying the values of $g_0$, $g_1$ and $g_2$. The values used are shown in Tab. 1, together with some characterization of the environments that would produce such parameter values.

---

[4]This space is spanned by tensor products of the 3 Pauli matrices and the $2 \times 2$ identity (this is effectively an MPO).

Table 1: Different sets of physical parameters used in the simulations. The dissipation parameters $g_0$, $g_1$ and $g_2$ are defined in Eqs. 7-9. The parameters $\alpha$, $\beta$ and $\gamma$ are linear combinations of the $g_i$ defined by Eq. 29.

|        | Environment                        | $g_0$ | $g_1$ | $g_2$ | $\alpha$ | $\beta$ | $\gamma$ |
|--------|------------------------------------|-------|-------|-------|----------|---------|----------|
| case 1 | Spontaneous emission only          | 1     | 0     | 0     | 0.5      | 1       | 1        |
| case 2 | Pure dephasing                     | 0     | 0     | 1     | 2        | 0       | 0        |
| case 3 | Low temperature, low dephasing     | 0.9   | 0.1   | 0.1   | 0.7      | 1       | 0.8      |
| case 4 | Generic dissipative rates          | 0.6   | 0.4   | 0.25  | 1        | 1       | 0.2      |
| case 5 | Infinite temperature with dephasing| 1     | 1     | 1     | 3        | 2       | 0        |

First, we look at Eq. 30 in the left panel of Fig. 1. The numerical results are in perfect agreement with the theory. Comparing the numerical data with the exact expression shows that the absolute value of the error is always below $10^{-9}$. In the following figures, the maximum error also never exceeds this value. Since the expectation value of a product of an odd number of $Y$ vanishes, $\langle YY \rangle$ is a *connected* correlation, and it shows a decay with a characteristic timescale $\sim 1/\alpha$.

Now we turn to Eq. 31, first in a simple case for the single qubit observable $\langle Z \rangle$. The result is displayed in the right panel of Fig. 1 (this corresponds to $n = 0$ and $l = 1$). The cases 2 and 5 are not shown since they have $\gamma = 0$ and thus $\langle Z \rangle = 0$. $\langle Z \rangle$ displays a relaxation toward the steady state value $\langle Z \rangle_{t \to \infty} = \frac{\gamma}{\beta} = \frac{g_0 - g_1}{g_0 + g_1}$. When $g_1 = 0$ this is simply a relaxation toward the $|0\rangle$ state. We also note that the effects of the correlations in the system are not visible in this observable, in the sense that the exact same behavior would be observed independently of the initial state provided that $\langle Z \rangle = 0$ on all qubits at time 0.

In the more complex case where $n \neq 0$, we cannot scale all the cases on one curve, so we chose to show the dependence in the number of qubits for one case. In the left panel of Fig. 2, we show the time evolution of $\langle YYZ \rangle$ (that is $n = 1$ and $l = 1$) for case 1 and different number of qubits. Again the simulations are in perfect agreement with the theory. This observable has a non-monotonous time evolution for a simple reason: from Eq. 31, we see that this three-point observable factorizes into $\langle YYZ \rangle = \langle YY \rangle \langle Z \rangle$, that is a product of a decreasing function

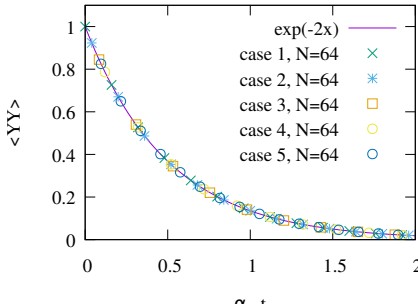
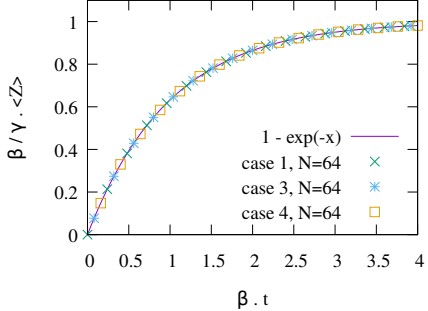

Figure 1: Left: Time evolution of $\langle YY \rangle$ in the different parameter cases with $N = 64$ qubits. The rescaled time $\alpha \cdot t$ in the horizontal axis allows the collapse of the curves associated to different sets of parameters. The line corresponds to Eq. 30 in the case $n = 1$. For this quantity, the maximum deviation between the numerical result and the exact value is less than $10^{-11}$. Right: same for $\langle Z \rangle$. For this quantity, the relevant rescaling of the time is $\beta \cdot t$. The line corresponds to Eq. 31 in the case $n = 0$ and $l = 1$. Cases 2 and 5 are not shown as they have $\gamma = 0$ and thus $\langle Z \rangle = 0$. The maximum deviation is here less than $10^{-9}$.

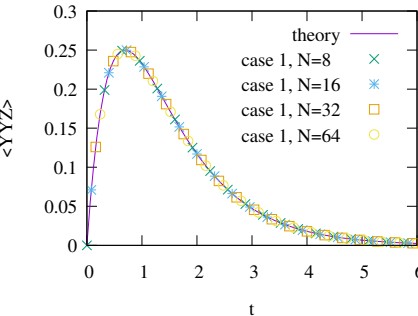 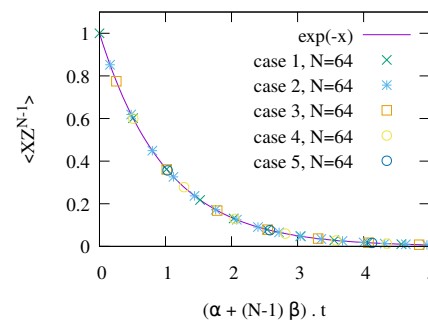

Figure 2: Left: Time evolution of $\langle YYZ\rangle$ for case 1 for different values of $N$. The line corresponds to Eq. 31 in the case $n = 1$ and $l = 1$. Cases 3 and 4 have similar behaviors, whereas cases 2 and 5 have $\gamma = 0$ and thus $\langle YYZ\rangle = 0$. As discussed in the text, the dynamics of this observable corresponds to the product of two exponentials, making it non-monotonous. The maximum deviation between numerics and theory is here less than $10^{-11}$. Right: Time evolution of the stabilizer $\langle XZ^{N-1}\rangle$. The rescaled time in the horizontal axis ensures that the different simulations (cases $1,\cdots,5$) fall onto the same curve, clearly showing the scaling of the $\beta$ contribution of the decay rate with the system size. The line corresponds to Eq. 32 for $n = 1$. The maximum deviation is here less than $10^{-12}$.

by an increasing function. Cases 3 and 4 have similar behaviors, whereas cases 2 and 5 have $\gamma = 0$ and thus $\langle YYZ\rangle = 0$.

Finally, we also checked the expectation value of the stabilizer of the complete graph state, that is Eq. 32 with $n = 1$. The results are displayed in the right panel of Fig. 2, they are again in perfect agreement with the theory. The initial value is 1, as it should be since the initial state is an eigenstate of the stabilizer for the eigenvalue 1. We then observe an exponential decay with a characteristic timescale given by $(\alpha + (N-1)\beta)^{-1}$ and which decreases with the number of qubits. This size-dependence of the decay rate can be viewed as a consequence of the fact that this specific observable involves all the qubits and reflects some global correlations in the system.

## 4.2 Operator space entanglement entropy

The method to compute the von Neumann entanglement entropy associated to a given bipartition of a graph state is explained in Ref. [4]. In the case of the complete graph state, the result is $S_{\mathrm{vN}} = \ln 2$ independent of the the bipartition (for two non-empty subsystems). This result is also easy to obtain using the fact that the complete graph state is LC-equivalent to the GHZ state.

For a mixed state $\rho$, it is interesting to consider the operator space entanglement entropy (OSEE) [37], a quantity that naturally arises in simulations of the density matrix dynamics represented using MPO. In a closely related context, entanglement entropies associated to (unitary) operators were also discussed in Ref. [38]. The OSEE has been the subject of many theoretical and numerical studies in the context of many-body problems. For instance, Ref. [39] considered the OSEE in thermal density matrices of spin chain models, Ref. [40] studied the OSEE in one dimensional many-body systems using (conformal) field theory techniques and [41] studied the long-time dynamics of OSEE in a spin chain subject to a dephasing dissipation (equivalent to the $g_2$ term in the present study). Another recent work [42] compared two different approaches for the dynamics of a dissipative quantum spin chain, Lindblad master equation versus quantum trajectories, and provided a comparison between the OSEE

in the Lindblad approach with the entanglement entropy in the trajectory formalism.

The OSEE can be defined by considering the vectorization $|\rho\rangle\rangle$ of $\rho$, which is a pure state in an enlarged Hilbert of dimension 4 per site (spanned by the 3 Pauli matrices plus the identity matrix). The OSEE associated to a given bipartition into two subsystems $A$ and $B$ is by definition the von Neumann entanglement entropy $\text{OSEE}^{A|B} = S_{\text{vN},|\rho\rangle\rangle}^{(A)} = S_{\text{vN},|\rho\rangle\rangle}^{(B)}$ associated to this partition of the vectorized pure state $|\rho\rangle\rangle$ (more details in App. A). The OSEE quantifies the total amount of correlations between the two subsystems. We note that the OSEE alone does not indicate whether the correlations are mostly classical or quantum.

For a pure state $|g\rangle$, the associated density matrix is $\rho = |g\rangle\langle g|$ and the OSEE of $\rho$ for a given bipartition is by construction twice the von Neumann entropy associated to the same bipartition in $|g\rangle$. So, in our model, the OSEE at time $t = 0$ is $2\ln 2$ for any nontrivial bipartition of the $N$ qubits. At long times, the system reaches a product state (if $g_1 = 0$ it simply corresponds to all the qubits in state $|0\rangle$) which is a state with bond dimension equal to 1 and vanishing OSEE. At intermediate finite times $t > 0$, the OSEE must therefore decrease from its initial value and eventually converge to 0 for any bipartition of the system. It is, however, no longer independent of the bipartition and in the rest of the paper, we will focus of the bipartition in two subsystems of equal size ($N/2$).

The dynamics of the OSEE for this bipartition in two equal halves of system is shown in Fig. 3. The interesting feature is the appearance of a very clear plateau at $\text{OSEE} = \ln 2$, whose duration grows with the number of qubits.

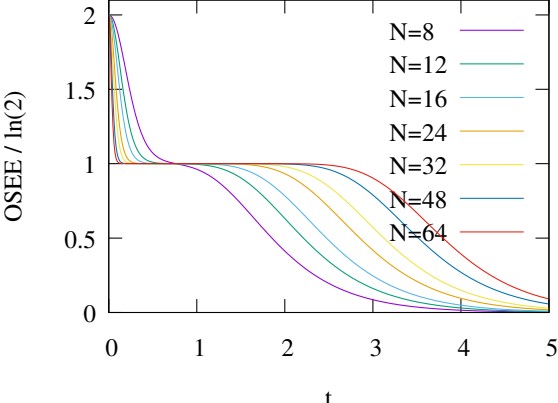

Figure 3: Time evolution of the OSEE between the two halves of the system for different number of qubits in case 1. A plateau in the OSEE value at intermediate times $t$, whose duration grows (logarithmically) with the number of qubits is clear.

To explore this behavior, we determined numerically the scaling laws for both the time at which the plateau begins and the time at which it ends. We observe a $1/N$ behavior at early times and $\ln N$ behavior for the time at the end of the OSEE plateau. The corresponding plots are shown in Fig. 4. At early times, the behavior is similar for cases 3, 4 and 5 with a $1/N$ behavior (left panel Fig. 4) and a plateau at $\text{OSEE} = \ln 2$. Case 2 is different with a plateau that starts at $t = 0$, see Fig. 5.

This behavior arises only in the case with $\beta = 0$, which corresponds to an absence of flip terms (pure dephasing). At late times, the $\ln N$ behavior is universal, with different values of the prefactor $\delta$ (up to some finite size effects the OSEE depends only on $t - \delta \ln N$ at late times). In case 2, the data collapse is particularly striking (see right panel of Fig. 5).

Comparing the values of $\delta$ to the parameters in each case, we see that for all our cases the value of $\delta$ is compatible with $\delta = 1/(2\alpha)$. It is remarkable that the OSEE stays essentially

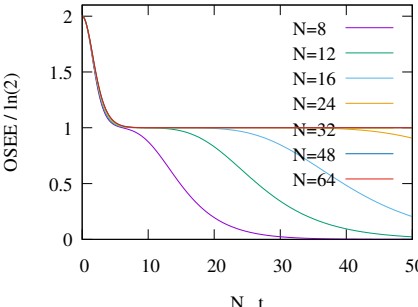 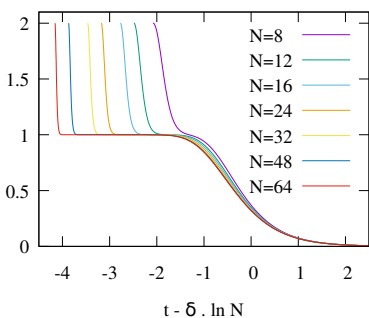

Figure 4: Time evolution of the OSEE between the two halves of the system for different values of $N$ in case 1 (same data as Fig. 3). Left: time rescaled by the system size $N$. The collapse of the curves at early times shows that the initial drop of the OSEE takes place over a timescale proportional to $1/N$. Right: same data with time shifted by $\delta \ln N$, here $\delta = 1$. In this panel the collapse of the curves at the end of the plateau illustrates the fact that the duration of the plateau is proportional to $\delta \ln N$.

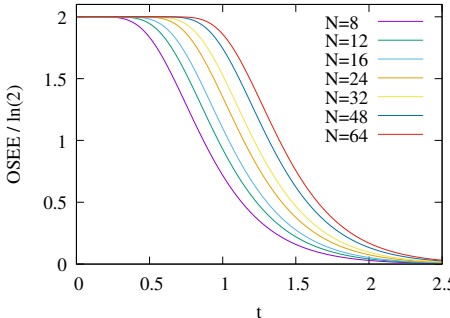 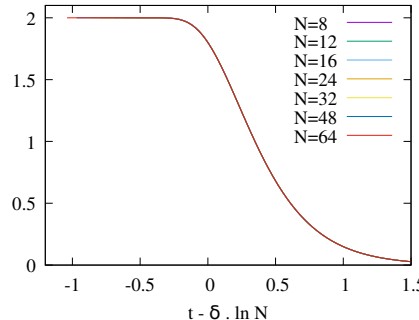

Figure 5: Time evolution of the OSEE between the two halves of the system for different number of qubits in case 2. Left: data plotted as a function of time. Right: data plotted as a function of time shifted by $\delta \ln N$, here $\delta = 0.25$. All the curves are essentially on top of one another and are indistinguishable at the scale of the figure.

constant for a duration that increases with the number of qubits, although the correlations decrease exponentially at best in constant time. A large enough system can be in a regime where $1/\beta \ll t$ and $t \lesssim \delta \ln(N)$. In case 1 ($g_0$ only), the first condition implies that $\langle Z \rangle$ is arbitrary close to 1, while the second condition puts the system in the OSEE plateau, with OSEE $\simeq \ln(2)$. In other words, the system can have an arbitrary low density of qubits in the $|1\rangle$ state and, still, some sizeable correlation between the two halves of the system.

We checked that such plateau is absent if the initial state is a graph state with a lower connectivity. As an example, Fig. 6 shows the OSEE in a case where the initial state is a graph state constructed from a periodic one-dimensional lattice with $N$ sites (a ring). For a subsystem $A$ of the form $A = [1, \cdots, n]$ with $N - 2 \geq n \geq 2$ the von Neumann entanglement entropy is equal to $S_{vN}^A = 2 \ln(2)$ in such a ring graph state [4], hence the value OSEE $= 4 \ln(2)$ at time $t = 0$. The correlations are plausibly only short-ranged (with a finite correlation length) in that case, so that the correlations between the two halves of the bipartition essentially come from the qubits close to the boundaries/cutting between the two halves. Thus, when the system size becomes significantly larger than the correlation length, the OSEE becomes independent of $N$. The OSEE then drops to zero over a characteristic timescale which is independent of the system size, contrary to the cases where the initial state is a complete graph.

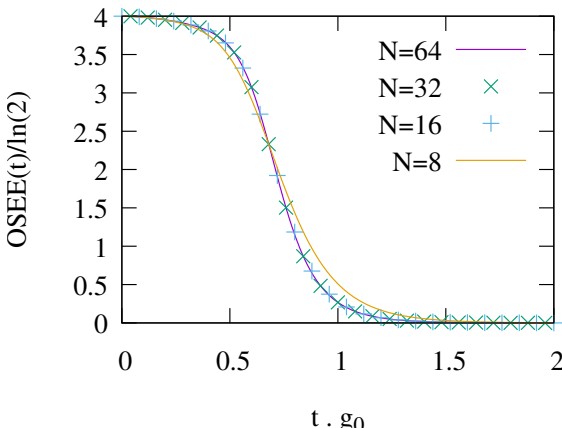

Figure 6: Time evolution of the OSEE between the two halves of the system, for a simulation of a *ring* graph state (initialized at $t = 0$), with the dissipator $\mathcal{D}_0$. The data for $N = 16$, 32 and 64 are almost on top of each other, showing rapid convergence to a limiting curve in the large $N$ limit. Contrary to the cases where the initial state is a complete graph, these curve do not exhibit any plateau. In this simulation the MPO bond dimension is 16 or below.

Finally, we also looked at what happens with a simple nonzero Hamiltonian. As an example we considered an Ising interaction. Namely, we chose $H = Z_a Z_b$ where $a$ and $b$ refer to two fixed qubits. When measuring the OSEE, one subsystem contained qubit $a$ and the other qubit $b$. With these parameters, the maximum bond dimension was 16. The results for case 1 are shown on Fig. 7. These figures are to be compared with Fig. 3 and 4 (right panel) where $H$ was zero. There are now four regimes: as before, at early times, a fast initial drop of the OSEE over a timescale proportional to $1/N$ (scaling not shown), at constant time (around $t \sim 0.8$) a peak, then a quasi plateau up to time $\delta \ln N$ and finally a decay to 0 in constant time ($\delta = 1$ in this case, as with $H = 0$). The peak is associated to a local OSEE maximum at time $t \sim 0.8$. We interpret this peak as a consequence of some new correlations created by the Ising term. These correlations are then washed out by the dissipation. We then have a regime of slow decay of the OSEE. In this regime the slope of the OSEE appears to scale as $1/\ln N$ at large $N$. As far as the OSEE is concerned, in this regime $H$ plays here the role of a perturbation to the plateau observed when $H = 0$, and the effect of this perturbation decreases when $N$ becomes large. Due to the observed finite-size scaling we expect a plateau to asymptotically appear in the large $N$ limit. However, to clearly separate the plateau from the peak one would need to go to much larger sizes, which is out of reach at the moment. We obtain similar results in the other cases (data not shown).

## 5   Conclusion

In this paper, we have introduced an exactly solvable toy model for the decoherence of a graph state. We have considered the time evolution of the complete graph state under a Lindblad equation with general single-qubit dissipators and no Hamiltonian. Exploiting the permutation invariance of the model has allowed obtaining simple closed formulae for the expectation values of any product of Pauli operators at any time. The method can be extended to other types of graph states, although the number of equations will grow for less symmetric situa-

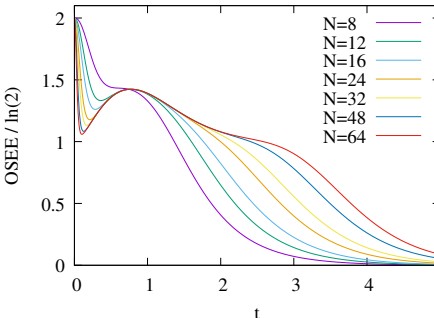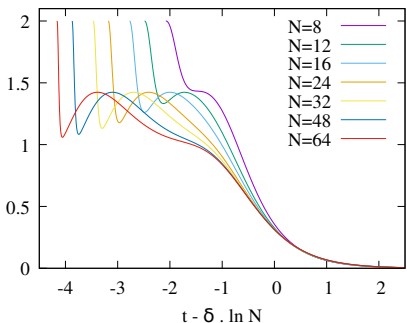

Figure 7: Left: Time evolution of the OSEE between the two halves of the system for different values of $N$ in case 1 with a nonzero Hamiltonian (a ZZ interaction between two qubits one in each half of the system). Right: same data with time shifted by $\delta \ln N$, here $\delta = 1$.

tions. The availability of analytic solutions is valuable as a guiding tool in understanding the dissipative mechanisms acting in numerical studies of complex setups.

We have compared the theoretical results with a numerical solver that was pushed up to 64 qubits. The results are in perfect agreement with the theory, showing that the MPO approach is adapted for simulating this type of problem.

A peculiar long-lasting plateau has been identified in the OSEE between the two halves of the system, pointing to the presence of some nonlocal long-lived correlations in this setup. Despite the dissipative processes acting everywhere on the qubits, the correlations have been observed to survive for a time that increases with the system size. The $t \sim \ln(N)$ scaling of the OSEE decay survives with a simple nonzero Hamiltonian. In a future study, we consider it valuable to compute exactly the OSEE for $t > 0$ in this model.

## Acknowledgments

G. M. thanks Élie Gouzien for useful discussions about graph states.

**Funding information** G. M. is supported by the PEPR integrated project EPiQ ANR-22-PETQ-0007 part of Plan France 2030.

## A  MPO representation of mixed state and operator space entanglement entropy

The density matrix can be expanded over all possible products of local Pauli operators:

$$\rho = \sum_{\alpha_1,\cdots,\alpha_N} C_{\alpha_1,\cdots,\alpha_N} \sigma^{\alpha_1} \otimes \cdots \sigma^{\alpha_N}, \tag{A.1}$$

with $\alpha_i \in \{0, x, y, z\}$ and $\sigma^0$ is the $2\times2$ identity matrix. In a vectorization picture, the operator above can be considered as a vector in the Hilbert space with dimension 4 at each site (spanned by tensor products of the 3 Pauli matrices and the $2 \times 2$ identity). It is then customary to use a super ket notation:

$$|\rho\rangle\rangle = \sum_{\alpha_1,\cdots,\alpha_N} C_{\alpha_1,\cdots,\alpha_N} |\sigma^{\alpha_1} \cdots \sigma^{\alpha_N}\rangle\rangle. \tag{A.2}$$

The MPO representation of $\rho$ is nothing but a matrix-product state (MPS) [43] description of the state above:

$$C_{\alpha_1,\cdots,\alpha_N} = \text{Tr}\left[M_1^{\alpha_1}\cdots M_N^{\alpha_N}\right].$$ (A.3)

If we choose large enough matrices $M_i^\alpha$, an arbitrary state can be represented in the above form. In particular, any state $\rho$ can be represented using matrices of size $\leq 4^N$. MPS/MPO representations however offer a computational advantage only when the state of interest can be accurately approximated using matrices of size $\ll 4^N$.

When representing a pure state $|\psi\rangle$ using an MPS, it is well known that an important quantity to consider is the so-called Schmidt spectrum, obtained by performing a singular value decomposition (SVD) of the state with respect to an $A|B$ bipartition of the system. This Schmidt spectrum is closely related to the von Neuman entropy associated to this bipartition.[5] Using the vectorization picture, the same idea applies to mixed state. Here also, what determines the matrix (or bond) dimension at a given position $n$ along the MPO (separating the left region $A$ with qubits $1,\cdots,n$ from the right region $B$ with qubits $n+1,\cdots,N$) is the Schmidt spectrum associated to the Schmidt decomposition of $|\rho\rangle\rangle$:

$$|\rho\rangle\rangle = \sum_k \lambda_k |\phi_k^A\rangle\rangle \otimes |\phi_k^B\rangle\rangle.$$ (A.4)

We usually normalize pure states with $\langle\psi|\psi\rangle = 1$. But for a mixed state the (Hilbert-Schmidt) square norm $\langle\langle\rho|\rho\rangle\rangle$ is generally different from unity since $\langle\langle\rho|\rho\rangle\rangle = \text{Tr}\left[\rho^2\right] = \sum_k \lambda_k^2$ is related to the purity of the state. By analogy with the von Neumann Entropy associated to the bipartition of a pure state, it is interesting to define an entropy associated to the Schmidt decomposition of a mixed state (Eq. A.4). To do so, we start by writing a normalized version of the state: $|\tilde\rho\rangle\rangle = \sum_k \tilde\lambda_k |\phi_k^A\rangle\rangle \otimes |\phi_k^B\rangle\rangle$ with rescaled Schmidt values $\tilde\lambda_k = \lambda_k/\left(\sum_l \lambda_l^2\right)^{1/2}$. The associated entropy, called operator space entanglement entropy [37, 38] is then defined by

$$\text{OSEE} = -\sum_k \tilde\lambda_k^2 \ln\left(\tilde\lambda_k^2\right).$$ (A.5)

To encode $\rho$ as an MPO, the bond dimension $\chi$ that one needs to use is closely related to the (exponential of the) OSEE, $\ln\chi \sim \text{OSEE}$.

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
