# Peer review of "A solvable model for graph state decoherence dynamics"

_SciPost Physics Core, doi:SciPost Phys. Core 7, 009 (2024)_

## Round 1 · Referee Report · Anonymous · 2023-8-18

Strengths

1) Analytic solution to open quantum system.
2) Benchmark against numerical method.
3) Additional example exploiting operator space entanglement entropy.

Weaknesses

1) Some of the general acceptance criteria are on the weak side, and considering them all together, they should improve in my opinion to make the paper stronger. I am referring here to the citations, some ambiguities in the language, and a detailed introduction. Some of the suggested points below address the primary concerns.
2) For reproducibility, the authors may think about attaching their setup for iTensor either as a data repository or as git repository, especially if they might consider the step in the future. Estimating the work behind it, I understand if they do not want to provide it, but it would fit well into SciPosts idea aiming at reproducibility.
3) Purely dissipative model without Hamiltonian, i.e., H=0.

Report

The article "A solvable model for graph state decoherence dynamics" (arXiv:2305.17231v1) describes the exactly solvable dynamics of a Lindblad master equation for graph states. The Lindblad master equation herein is purely dissipative and the results are scalable to the many-body regime. Therefore, the simulations are a useful fixpoint to compare with numerical simulations like tensor network methods. The exactly solvable model and the comparison to tensor network methods are the strong points of the article. I would suggest some revisions concerning the presentation before being accepted for publication in Scipost Physics Core; then, I would be looking forward to seeing it published in SciPost Physics Core. Larger revisions including cases with a Hamiltonian would move the submission probably into the top 50% in Scipost Physics Core.

Requested changes

Suggested revisions before acceptance:

The following points 1 - 12 are the suggested revisions; if they cannot be resolved, a comment why the point does not apply in the response to the referee report would be appreciated.

1) The case with a Hamiltonian unequal to zero would unlock a lot of potential use cases. It is understandable if the Hamiltonian cannot be added in the revision, although it would be highly beneficial to the paper, e.g., as a simple Ising model or XXZ model. If H is not added, I am missing a detailed comment if the authors consider it unfeasible or which complications they anticipate or already encountered when trying to integrate H into the model.
1a) page 6, "thermal steady state population": the thermal state goes a bit against the H=0 choice, so I would maybe clarify that this temperature requires choosing a diagonal local Hamiltonian and choosing the coupling of the Lindblad operators according to the temperature and energy gap.
1b) The conclusion is talking about a "vanishing Hamiltonian", while the main part does only take into account H = 0. I would therefore replace "a vanishing" with "no".

2) The citations are a bit sparse and could improve, especially after the introduction. It would benefit the reader to repeat relevant citations in the main part and give a broader view of the literature in each chapter. More citations would align as well with the suggested citation policy on SciPost.
2a) One example would be the section on tensor network methods, where the complete simulation side is covered with few citations. In my opinion, it should be extended in the introduction and the chapter about numerical methods.
2b) The operator space entanglement entropy is covered basically with one citation, where the field is probably much bigger. The work of P. Zanardi would be something to mention, likely amongst others. The OSEE is not mentioned in the introduction at all, although it covers a complete subchapter later.
2c) It is good practice to cite software and I am glad to see the authors do so, see Ref [21, 22]. Can the authors add the developers in Ref [21] and ideally the version or commit with which they were running the simulations? This information would add valuable information for reproducibility.
2d) Names are frequently not capitalized in the citations, e.g., it should be Greenberger-Horne-Zeilinger; please correct.

3) Page 6, "It is interesting, ... inversely proportional to the system size": as the analogy to the GHZ is frequently mentioned, this feature looks less surprising in my opinion. Also, the coherence in the GHZ state depends on the number of qubits as decay or dephasing on any qubit will affect the GHZ state. Therefore it would benefit the article to investigate if it is linked to the number of neighbors in the initial graph by running additional numerical simulations or if it comes from the similarity to the GHZ state. The language is a bit vague here ("may be related"; "interesting", but interesting because surprising, contradicting or confirming previous knowledge) and could improve.

4) Page 7, "presumably": the language is vague here, does it hold or not? Up to which system size do the authors have numerical evidence? Can it be constructed as well from a sum of outer products, which could help to prove separability? Defining Mz = [[z+, 0], [0, z-]] and My = [[0, -i y], [i y, 0]] and x as outer / Kronecker product, rho_2 = 0.25 * Mz x Mz + 0.25 * My x My which makes the argument about being a separable state easier. Is there maybe a similar construction for larger reduced density matrices or is by "product" already meant outer product.

5) Page 7,8: The authors point out the perfect agreement between numerical simulations and theory, but on the plots, one can probably only say that the errors are less than five to ten percent. Can the authors please add an error plot of the absolute difference between the theory results and the numerical results on a log-scale? Log-error plots are an essential step of a numerical analysis in my opinion. These plots would also fit well into an extended appendix.

6) Page 8, caption of Table 1: Captions should aim to be self-consistent and this caption neither explains g0, g1, g2 nor alpha, beta, gamma.

7) Page 8, Fig 1.: A comment on the left frame about why all cases yield the same expectation value would be useful. Is it expected? At this stage of the article, it raises the question of why all the cases were defined, which only becomes clear later on.

8) Page 10, caption Fig. 4: "by a factor N" sounds like a new variable, unknown to the reader. Since N is the system and assuming the factor is actually the system size, it should read "by the system size N". In contrast, if it is a factor different from the system size, a new variable should be used instead of N.

9) Page 11: "suppose" and "seems" could be more clear (first paragraph on page 11). What are the indications that it holds or does not hold?

10) Page 12: Discussion about MPO representation beyond 1d, where the claim is that the simulation indicates that MPOs work in "a high space dimension", which seems to be beyond 1d. This statement is a bit misleading as the GHZ-like states can be prepared in 1d systems and are not limited to 2d or higher dimensional systems. Furthermore, I would characterize the dimensionality of the system by the interactions present in the system, where we have only local Lindblad operators and therefore no obvious geometry. The dimensionality might affect the efficiency of how the initial state can be prepared, but does not affect the time evolution discussed in the article in my opinion. As pointed out in one of the next comments, the initial bond dimension should be sufficient for local problems and therefore I would remove the whole statement about suitability for higher dimensions unless it is reflected in the propagator when changing the dimensionality.

11) Page 12, "possibly classical": vague language, as a reader I would like to know if they are classical or not, e.g., for readers interested in using quantum correlations as a resource. Even if the correlator to be considered is not known, tools like quantum mutual information should help to distinguish classical from quantum correlations and make a clear statement here.

12) Page 13: Bond dimension and small numerical errors: the bond dimension of the MPO of the density matrix is known for the two limits, i.e., the initial state and the steady state. As we have a local model without any interaction terms, the bond dimension should not grow beyond the bond dimension of the initial state, i.e., 4. What kind of update do the authors use for their numerical simulation? A two-tensor update or a single-tensor update? Are there global Krylov steps acting on the whole state? Knowing that the evolution operator is local, a single-tensor update is sufficient in my opinion, and would probably eliminate the numerical errors.

Optional revisions (no need to respond if not taken into consideration)

13) The authors could add some structural elements, which are not strictly necessary, but might improve the presentation.
13a) The structure of the paper does not appear at the end of the introduction, but at the end of Sec. 2. I would move it to the introduction unless it contradicts a SciPost policy.
13b) To avoid empty sections between a section and a subsection, e.g., see Sec. 3 and Sec. 3.1. or 4 and 4.1, it would be beneficial to add one short paragraph to explain to the reader what follows in the complete section.

14) Usage of past tense
14a) In my opinion, inconsistent use of simple past and present perfect in the conclusion, i.e., "we introduced" (p. 11) vs. "have compared" (p. 12). It should be consistent and, in my opinion, present perfect in the conclusion.
14b) The main part switches between simple past and present, e.g., on page 8 with "we also checked" and "we then observe", but I assume that was intended by the authors.

15) For the appendix, there are several small comments:
15a) The appendix with one paragraph seems to fit into the main part as is or should be extended, e.g., with the convergence study suggested in a previous point.
15b) A small paragraph with two sentences to introduce what the appendix discusses would be helpful before starting with details.
15c) What criteria do the simulations use in terms of cutting singular values? The bond dimension encountered as 15 will depend on which singular values are cut, e.g., if one cuts everything below 1e-8 or 1e-12. Please mention this parameter as well.

16) Page 3, Eq. (6): maybe a mention of the possibility to map it to a vector, i.e., super-ket, is here already useful. It is used for many numerical methods and allows readers to connect. Furthermore, it is used later on.

17) Page 4: * in the subscript is not explained, e.g., with "where $* \in \{0, 1, 2 \}$".

18) Page 4: "nontrivial action" to "non-zero" action?

19) Page 4: replace "is the same" with "has the same expectation value"

20) page 5, Eq. (18): can the authors highlight somehow how many qubits are in the state $\ket{-}$, maybe as $\ket{- \ldots -}_{1 \ldots n} \ket{+ \ldots +}_{n+1 \ldots N}$?

21) page 5: The language could be stronger here, e.g., multiple "do" and "this" without explicit reference to what "this" refers. For example after Eq. (19), "To be real, the last expression of Eq. (19) must be 0 ..." would be more accessible to the reader.

22) Page 6: "gives the two terms" to "results in two terms"

23) Page 6, "recurrence": would "recursion" be even more fitting as a term here?

24) Page 7: consistency, it is "2-qubit density matrix", but "two-point" correlations. Usually, small numbers can be written out as words.

25) Page 7, "whatever": replace with "independent of the" (whatever sounds colloquial)

26) Page 8: replace "whatever" with "independent of" (whatever sounds colloquial)

27) Page 8: "as it should" to "as it should be"

28) Page 8, caption Fig. 1. To aim at a self-consistent caption, I suggest writing "cases with N=64 qubits" to explain the variable N.

29) Page 9: "interesting", but interesting because surprising, contradicting or confirming previous knowledge?

30) Page 10, caption Fig. 3. To aim at a self-consistent caption, I suggest specifying "at intermediate times $t$" to explain the x-axis label.

31) Page 11: add a comma to ease the flow of reading: "... larger than the correlation length, the OSEE becomes ..."

32) Page 12: multiple uses of conditional, i.e., "would". I suggest "of analytic solutions is valuable" and "In a future study, we consider it valuable to compute".

---

## Round 1 · Referee Report · Anonymous · 2023-8-31

Strengths

- The manuscript provide an analytical solution to a non-trivial open many-body problem
- Interesting insights for the behavior of the OSEE

Weaknesses

- The authors consider only the H=0 case

Report

The authors of the manuscript "A solvable model for graph state decoherence dynamics" an exact analytical solution of the dissipative-only dynamics of permutation-invariant graph states of N qubits. They provide the exact equation of motions for observables as well as predictions on the OSEE. Analytical predictions are benchmarked against tensor-network simulations and shown to be exact. The paper is very pedagogical and allows the reader to follow all the calculations that has been done. The theoretical framework is robust and scientifically sound. I support the publication of the manuscript provided that the authors answer to my criticisms.

Requested changes

- On the reduced density matrices. In Sec.3.4 the authors provide the structure of the reduced 2- and 3-site density matrix. They comment on the fact \rho_2 is separable at all times while \rho_3 and \rho_4 show zero negativity. Given that the construction of the N-qubit density matrix can be obtained by iteratively I was wondering if the global density matrix of the N qubits can be constructed as well. If this is the case would be possible to obtain some results for the time evolution of the system negativity at least for small N?

- Again on entanglement. I understand that the OSEE does not distinguish between classical and quantum correlations. As a matter of fact the negativity is entanglement witness. For small N it would be nice to compare the behavior of the entanglement negativity compared to the OSEE and see how classical and quantum correlation are progressively killed by the dissipative evolution.

---

## Round 2 · Referee Report · Anonymous (Referee 1) · 2024-1-17

Strengths

1) Analytic solution to open quantum system. 2) Benchmark against numerical method. 3) Additional example exploiting operator space entanglement entropy.

Weaknesses

The weaknesses stated in the Report-1 have been improved by the recent revision. No more weaknesses to report here remain.

Report

I recommend proceeding with publication in SciPost Physics Core. The authors have addressed all points from the initial reports, which improved the manuscript.

---

## Round 2 · Referee Report · Anonymous (Referee 2) · 2024-1-30

Report

I read the reply of the authors to the referee reports as well as the new version of the manuscript. The authors have addressed all my concerns and therefore I support the publication of the manuscript in its present form.

---

## Round 2 · Author Response

We thank both referees for their very careful analysis of the manuscript and for their numerous useful comments and questions. We have essentially taken into account all their suggestions (see the list of changes below).

---

## Round 2 · List of Changes

*** Answers to Referee #1 ***

1) The case with a Hamiltonian unequal to zero would unlock a lot of potential use cases. It is understandable if the Hamiltonian cannot be added in the revision, although it would be highly beneficial to the paper, e.g., as a simple Ising model or XXZ model. If H is not added, I am missing a detailed comment if the authors consider it unfeasible or which complications they anticipate or already encountered when trying to integrate H into the model.

We have looked at a simple case with a nonzero H. As suggested, we considered an Ising interaction acting on (only) two qubits. The (new) figure 7 displays the OSEE in that case, and it shows that the t~ln(N) scaling of the OSEE decay persists. A paragraph has been added at the end of Sec. 4.2

1a) page 6, "thermal steady state population": the thermal state goes a bit against the H=0 choice, so I would maybe clarify that this temperature requires choosing a diagonal local Hamiltonian and choosing the coupling of the Lindblad operators according to the temperature and energy gap.

In order to clarify this point we have added the following footnote: In the long time limit and for $n=0$ and $l=1$ the Eq.~\ref{eq_ev_yyz} below gives $\langle Z \rangle_{t\gg\beta}= \frac{\gamma}{\beta}$ which is the thermal equilibrium value if we introduce a local Hamiltonian $H=\Delta \sum_i Z_i$ and a temperature $T$ satisfying $\tanh(\Delta/(k_B T))=\frac{\gamma}{\beta}$.

1b) The conclusion is talking about a "vanishing Hamiltonian", while the main part does only take into account H = 0. I would therefore replace "a vanishing" with "no".

This as been corrected.

2) The citations are a bit sparse and could improve, especially after the introduction. It would benefit the reader to repeat relevant citations in the main part and give a broader view of the literature in each chapter. More citations would align as well with the suggested citation policy on SciPost. 2a) One example would be the section on tensor network methods, where the complete simulation side is covered with few citations. In my opinion, it should be extended in the introduction and the chapter about numerical methods.

We added five new references in the introduction in relation to the creation and characterization of GHZ states with various quantum hardware systems.

We added several new references about tensor network methods at the beginning of Section 4:

  • T. Prosen and M. Žnidariˇc, J. Stat. Mech. 2009(02), P02035 (2009)
  • F. Verstraete, J. J. García-Ripoll and J. I. Cirac, Phys. Rev. Lett. 93(20), 207204 (2004)
  • M. Zwolak and G. Vidal, Phys. Rev. Lett. 93(20), 207205 (2004)
  • M. P. Zaletel et al. , Phys. Rev. B 91(16), 165112 (2015)
  • K. Bidzhiev and G. Misguich, Phys. Rev. B 96(19), 195117 (2017)

We also added a new appendix ("MPO representation of mixed state and Operator Space Entanglement Entropy") with another additional reference:

  • U. Schollwöck, Annals of Physics 326(1), 96 (2011)
  • N. Schuch et al. Phys. Rev. Lett. 100(3), 030504 (2008)

2b) The operator space entanglement entropy is covered basically with one citation, where the field is probably much bigger. The work of P. Zanardi would be something to mention, likely amongst others. The OSEE is not mentioned in the introduction at all, although it covers a complete subchapter later.

On the OSEE topic we have added the following references: - P. Zanardi, Phys. Rev. A 63(4), 040304 (2001) - M. Žnidariˇc, T. Prosen and I. Pižorn, Phys. Rev. A 78(2), 022103 (2008) - J. Dubail, J. Phys. A: Math. Theor. 50(23), 234001 (2017) - D. Wellnitz, G. Preisser, V. Alba, J. Dubail and J. Schachenmayer, Phys. Rev. Lett. 129(17), 170401 (2022) - G. Preisser, D. Wellnitz, T. Botzung and J. Schachenmayer, Phys. Rev. A 108(1), 012616 (2023)

2c) It is good practice to cite software and I am glad to see the authors do so, see Ref [21, 22]. Can the authors add the developers in Ref [21] and ideally the version or commit with which they were running the simulations? This information would add valuable information for reproducibility.

The (main) authors of the software are the authors of Ref[21] and we have merged Refs 21 and 22.

2d) Names are frequently not capitalized in the citations, e.g., it should be Greenberger-Horne-Zeilinger; please correct.

This has been fixed.

3) Page 6, "It is interesting, ... inversely proportional to the system size": as the analogy to the GHZ is frequently mentioned, this feature looks less surprising in my opinion. Also, the coherence in the GHZ state depends on the number of qubits as decay or dephasing on any qubit will affect the GHZ state. Therefore it would benefit the article to investigate if it is linked to the number of neighbors in the initial graph by running additional numerical simulations or if it comes from the similarity to the GHZ state. The language is a bit vague here ("may be related"; "interesting", but interesting because surprising, contradicting or confirming previous knowledge) and could improve.

The analysis of more general initial graph states shows that this decay rate is indeed proportional to the number of neighbors in the initial graph. We have modified the text accordingly.

4) Page 7, "presumably": the language is vague here, does it hold or not? Up to which system size do the authors have numerical evidence? Can it be constructed as well from a sum of outer products, which could help to prove separability? Defining Mz = [[z+, 0], [0, z-]] and My = [[0, -i y], [i y, 0]] and x as outer / Kronecker product, rho_2 = 0.25 * Mz x Mz + 0.25 * My x My which makes the argument about being a separable state easier. Is there maybe a similar construction for larger reduced density matrices or is by "product" already meant outer product.

We have modified the corresponding text. In fact, since the reduced density matrices are separable at the initial time (known results for GHZ states) and since the evolution is only due to single-qubit dissipators, the reduced density matrices indeed remain separable at all times.

5) Page 7,8: The authors point out the perfect agreement between numerical simulations and theory, but on the plots, one can probably only say that the errors are less than five to ten percent. Can the authors please add an error plot of the absolute difference between the theory results and the numerical results on a log-scale? Log-error plots are an essential step of a numerical analysis in my opinion. These plots would also fit well into an extended appendix.

We have measured the errors associated to all the data plotted in the manuscript (which is straightforward since we have the exact formula in all cases). The result is the errors (in absolute value) are always below 10^{-9}, and in fact most of the time even smaller (10^{-11}). Because these values are so tiny adding plots showing these errors would not add much information. We have however specified the magnitude of these errors in the text and in the figure captions.

6) Page 8, caption of Table 1: Captions should aim to be self-consistent and this caption neither explains g0, g1, g2 nor alpha, beta, gamma.

The caption of Table 1 has been modified, it now includes the numbers of the equations where g0, g1, g2 nor alpha, beta and gamma are defined.

7) Page 8, Fig 1.: A comment on the left frame about why all cases yield the same expectation value would be useful. Is it expected? At this stage of the article, it raises the question of why all the cases were defined, which only becomes clear later on.

Concerning the observable <YY> displayed the left panel, the exact expression was computed before in the text (Eq. 30). This expression shows that the physical parameters only appear through the combination alphat Therefore it was already clear from there that the curves associated to the different cases would collapse when plotted as a function of alphat. So, this was indeed expected.

8) Page 10, caption Fig. 4: "by a factor N" sounds like a new variable, unknown to the reader. Since N is the system and assuming the factor is actually the system size, it should read "by the system size N". In contrast, if it is a factor different from the system size, a new variable should be used instead of N.

Corrected

9) Page 11: "suppose" and "seems" could be more clear (first paragraph on page 11). What are the indications that it holds or does not hold?

This plateau is indeed present if and only if beta=0. We have made this more precise in the text.

10) Page 12: Discussion about MPO representation beyond 1d, where the claim is that the simulation indicates that MPOs work in "a high space dimension", which seems to be beyond 1d. This statement is a bit misleading as the GHZ-like states can be prepared in 1d systems and are not limited to 2d or higher dimensional systems. Furthermore, I would characterize the dimensionality of the system by the interactions present in the system, where we have only local Lindblad operators and therefore no obvious geometry. The dimensionality might affect the efficiency of how the initial state can be prepared, but does not affect the time evolution discussed in the article in my opinion. As pointed out in one of the next comments, the initial bond dimension should be sufficient for local problems and therefore I would remove the whole statement about suitability for higher dimensions unless it is reflected in the propagator when changing the dimensionality.

We agree with the referee and the statement has been removed.

11) Page 12, "possibly classical": vague language, as a reader I would like to know if they are classical or not, e.g., for readers interested in using quantum correlations as a resource. Even if the correlator to be considered is not known, tools like quantum mutual information should help to distinguish classical from quantum correlations and make a clear statement here.

We have removed from the conclusion the expression "possibly classical". The correlations responsible for the plateau could be 1) classical or 2) quantum but involving the whole system. In the scenario 2) the fact that the correlation must involve the whole system is due to the fact there is no entanglement involving strictly less than the N qubits.

Concerning the use of quantum mutual information: as far as we know, in mixed states, this quantity can be nonzero even for separable (i.e. classical) states. It is therefore not clear to us how this tool could be used here.

12) Page 13: Bond dimension and small numerical errors: the bond dimension of the MPO of the density matrix is known for the two limits, i.e., the initial state and the steady state. As we have a local model without any interaction terms, the bond dimension should not grow beyond the bond dimension of the initial state, i.e., 4. What kind of update do the authors use for their numerical simulation? A two-tensor update or a single-tensor update? Are there global Krylov steps acting on the whole state? Knowing that the evolution operator is local, a single-tensor update is sufficient in my opinion, and would probably eliminate the numerical errors.

We use two-tensor updates (which is generically more powerful) and we do not make use Krylov steps. Note that even if we observe some bond dimension increase the errors on observables are extremely small (10^{-9} or below).

-- Optional revisions (no need to respond if not taken into consideration)

13) The authors could add some structural elements, which are not strictly necessary, but might improve the presentation. 13a) The structure of the paper does not appear at the end of the introduction, but at the end of Sec. 2. I would move it to the introduction unless it contradicts a SciPost policy.

Corrected

13b) To avoid empty sections between a section and a subsection, e.g., see Sec. 3 and Sec. 3.1. or 4 and 4.1, it would be beneficial to add one short paragraph to explain to the reader what follows in the complete section.

Done

14) Usage of past tense 14a) In my opinion, inconsistent use of simple past and present perfect in the conclusion, i.e., "we introduced" (p. 11) vs. "have compared" (p. 12). It should be consistent and, in my opinion, present perfect in the conclusion.

Done

14b) The main part switches between simple past and present, e.g., on page 8 with "we also checked" and "we then observe", but I assume that was intended by the authors.

15) For the appendix, there are several small comments: 15a) The appendix with one paragraph seems to fit into the main part as is or should be extended, e.g., with the convergence study suggested in a previous point.

This short appendix has been removed, and the associated text has been dispatched at different places in the text.

15b) A small paragraph with two sentences to introduce what the appendix discusses would be helpful before starting with details.

This short appendix has been removed.

15c) What criteria do the simulations use in terms of cutting singular values? The bond dimension encountered as 15 will depend on which singular values are cut, e.g., if one cuts everything below 1e-8 or 1e-12. Please mention this parameter as well.

The cut off parameter has been added in the text (10^{-16}).

16) Page 3, Eq. (6): maybe a mention of the possibility to map it to a vector, i.e., super-ket, is here already useful. It is used for many numerical methods and allows readers to connect. Furthermore, it is used later on.

This has been explained in the (new) appendix A.

21) page 5: The language could be stronger here, e.g., multiple "do" and "this" without explicit reference to what "this" refers. For example after Eq. (19), "To be real, the last expression of Eq. (19) must be 0 ..." would be more accessible to the reader.

corrected

22) Page 6: "gives the two terms" to "results in two terms"

corrected

23) Page 6, "recurrence": would "recursion" be even more fitting as a term here?

corrected

24) Page 7: consistency, it is "2-qubit density matrix", but "two-point" correlations. Usually, small numbers can be written out as words.

corrected

25) Page 7, "whatever": replace with "independent of the" (whatever sounds colloquial)

corrected

26) Page 8: replace "whatever" with "independent of" (whatever sounds colloquial)

corrected

27) Page 8: "as it should" to "as it should be"

corrected

28) Page 8, caption Fig. 1. To aim at a self-consistent caption, I suggest writing "cases with N=64 qubits" to explain the variable N.

corrected

29) Page 9: "interesting", but interesting because surprising, contradicting or confirming previous knowledge?

30) Page 10, caption Fig. 3. To aim at a self-consistent caption, I suggest specifying "at intermediate times t" to explain the x-axis label.

corrected

31) Page 11: add a comma to ease the flow of reading: "... larger than the correlation length, the OSEE becomes ..."

corrected

32) Page 12: multiple uses of conditional, i.e., "would". I suggest "of analytic solutions is valuable" and "In a future study, we consider it valuable to compute".

corrected

*** Answers to Referee #2 ***

1) On the reduced density matrices. In Sec.3.4 the authors provide the structure of the reduced 2- and 3-site density matrix. They comment on the fact \rho_2 is separable at all times while \rho_3 and \rho_4 show zero negativity. Given that the construction of the N-qubit density matrix can be obtained by iteratively I was wondering if the global density matrix of the N qubits can be constructed as well. If this is the case would be possible to obtain some results for the time evolution of the system negativity at least for small N?

Sec. 3.4 has been improved. We know that the initial state density matrices rho_k are separable (provided that k<N) at the initial state (simply by importing a known result on the GHZ state). Then, from the fact that the time evolution is generated by some single qubit dissipators (which cannot create entanglement), it is in fact clear that the density matrices rho_[k<N} must remain separable at all times in the present model.

2) Again on entanglement. I understand that the OSEE does not distinguish between classical and quantum correlations. As a matter of fact the negativity is entanglement witness. For small N it would be nice to compare the behavior of the entanglement negativity compared to the OSEE and see how classical and quantum correlation are progressively killed by the dissipative evolution.

For (very) small N it is indeed conceivable to get an exact expression for the density matrix of the entire system. The negativity associated to a given partition could then be plotted as a function of time and compared with the OSEE of the same bi-partition. Note however that the two quantities may a priori scale differently with the system size. We have not yet undertaken this calculation but it is certainly an interesting direction to explore.

---

## Editorial Decision

published